# MAGE-A10 Protein Expression in Advanced High Grade Serous Ovarian Cancer Is Associated with Resistance to First-Line Platinum-Based Chemotherapy

**DOI:** 10.3390/cancers15194697

**Published:** 2023-09-23

**Authors:** Nataša Lisica Šikić, Branka Petrić Miše, Snježana Tomić, Giulia Spagnol, Luka Matak, Antonio Juretić, Giulio Spagnoli

**Affiliations:** 1Department of Pathology, Forensic Medicine and Cytology, General Hospital Zadar, 23000 Zadar, Croatia; nlisicasikic@gmail.com; 2Department of Oncology, Clinical Hospital Center Split, School of Medicine, University of Split, 21000 Split, Croatia; brapemi@gmail.com; 3Department of Pathology, Forensic Medicine and Cytology, Clinical Hospital Center Split, School of Medicine, University of Split, 21000 Split, Croatia; snjezana.tomic17@gmail.com; 4Department of Women and Children’s Health, Clinic of Gynecology and Obstetrics, University of Padua, 35122 Padua, Italy; giuliaspagnol.ts@gmail.com; 5Department of Obstetrics and Gynecology, General Hospital Zadar, 23000 Zadar, Croatia; 6Department of Oncology, University Hospital Dubrava, University of Zagreb, 10000 Zagreb, Croatia; ajuretic@kbd.hr; 7Istituto CNR “Translational Pharmacology”, 00133 Rome, Italy; gcspagnoli@gmail.com

**Keywords:** ovarian cancer, MAGE- A10, NY-ESO-1, immunohistochemistry, response to platinum-based chemotherapy, platinum sensitivity, prognosis

## Abstract

**Simple Summary:**

The Melanoma Antigen Gene (MAGE) protein family is a large group of proteins that share a common MAGE homology domain. Many MAGE proteins are aberrantly expressed in a wide variety of cancer types and are of interest as a biomarkers in cancer and targets of immunotherapies because of a subset of the proteins that are classified as cancer/testis antigens (CTAs). In ovarian cancers, MAGE-A1, -A9, and -A10 expression are associated with worse prognosis and unresponsiveness to platinum-based chemotherapy.

**Abstract:**

Ovarian cancer has a dismal prognosis. Standard treatment following surgery relies on platinum-based chemotherapy. However, sizeable percentages of patients are unresponsive. Identification of markers predicting the response to chemotherapy might help select eligible patients and spare non-responding patients from treatment-associated toxicity. Cancer/testis antigens (CTAs) are expressed by healthy germ cells and malignant cells of diverse histological origin. This expression profile identifies them as attractive targets for cancer immunotherapies. We analyzed the correlations between expression of MAGE-A10 and New York esophageal-1 cancer (NY-ESO-1) CTAs at the protein level and the effectiveness of platinum-based chemotherapy in patients with advanced-stage high-grade serous ovarian carcinoma (HGSOC). MAGE-A10 and NY-ESO-1 protein expression was analyzed by immunohistochemistry (IHC) in formalin-fixed, paraffin-embedded samples from 93 patients with advanced-stage HGSOC treated at our institutions between January 1996 and December 2013. The correlation between the expression of these markers and response to platinum-based chemotherapy, evaluated according to RECIST 1.1 criteria and platinum sensitivity, measured as platinum-free interval (PFI), progression free (PFS), and overall survival (OS) was explored. The MAGE-A10 protein expression predicted unresponsiveness to platinum-based chemotherapy (*p* = 0.005), poor platinum sensitivity (*p* < 0.001), poor PFS (*p* < 0.001), and OS (*p* < 0.001). Multivariate analysis identified MAGE-A10 protein expression as an independent predictor of poor platinum sensitivity (*p* = 0.005) and shorter OS (*p* < 0.001). Instead, no correlation was observed between the NY-ESO-1 protein expression and response to platinum-based chemotherapy (*p* = 0.832), platinum sensitivity (*p* = 0.168), PFS (*p* = 0.126), and OS (*p* = 0.335). The MAGE-A10 protein expression reliably identified advanced-stage HGSOC unresponsive to platinum-based chemotherapy. Targeted immunotherapy could represent an important alternative therapeutic option in these cancers.

## 1. Introduction

Ovarian cancer is the most lethal of all gynecologic malignant tumors. It is often diagnosed at a late stage and the 5-year overall survival (OS) rate is around 45% [1].

The international standard of care for women with advanced ovarian cancer is represented by surgical resection, which is usually followed by platinum-based chemotherapy. Platinum-based drug combinations are also included in second-line chemotherapy protocols for the treatment of patients with platinum-sensitive, but relapsing ovarian cancers [2].

However, in 20–40% of patients platinum-based chemotherapy is ineffective [2] and salvage chemotherapy administered to these patients is also frequently characterized by low response rates.

A multiplicity of molecular mechanisms have been suggested to underlie resistance to platinum-based treatment. They might include altered expression of platinum transporter proteins, overexpression of detoxificating compounds, and enhanced DNA repair processes [3].

On the other hand, administration of platinum-based treatments has widely been shown to be frequently accompanied by severe side effects, including nephrotoxicity, neurotoxicity, and, most importantly, myelosuppression, resulting in anemia, reduction of platelet counts, and impaired immune responses [4].

Identification of markers of potential clinical use predicting resistance to platinum-based chemotherapy is urgently needed, to spare unnecessary toxicity to patients with unresponsive cancers, and to envisage alternative treatments of these malignancies.

In previous studies [5], we have shown that high E-cadherin expression, as detectable by standard immunohistochemistry (IHC) techniques, reliably identifies advanced high-grade serous ovarian cancers (HGSOCs) responding to platinum-based chemotherapy. These results critically contribute to the selection of patients taking advantage of these treatments, but fail to provide clues favoring the development of alternative therapies of potential interest for unresponsive patients.

Notably, immunotherapies based on immunological checkpoint blockade (ICB) are known to be poorly effective in HGSOC [6], possibly due to a relatively low mutational burden [7], resulting in a limited generation of neo-antigens [8].

More recently, adoptive immunotherapies, based on the transfer of gene-engineered T cells expressing chimeric antigen receptors (CARs) have been proposed for the treatment of advanced ovarian cancers and associated peritoneal carcinomatosis [9]. However, an important obstacle for the development of adequate protocols is represented by the relative paucity of tumor “specific” markers expressed by HGSOC cells, significantly increasing the risk of “on target, off tumor” side effects [9].

Cancer/testis antigens (CTAs) are a family of proteins highly expressed in germinal cells and cancers of different histological origin [10,11,12] including ovarian cancers [13].

Members of the melanoma-associated antigen-A (MAGE-A) CTA subfamily are overexpressed in many cancers and have been suggested to be involved in tumor progression, metastasis, and resistance to treatment, and to be associated with poor prognosis also in ovarian cancers [13,14]. Remarkably, MAGE-A10 is one of the most immunogenic CTAs, and specific cellular immune responses have been observed in the peripheral blood of healthy donors and patients with cancer [15,16].

New York esophageal squamous cell carcinoma-1 (NY-ESO-1) is another well-characterized CTA, expressed in numerous malignancies including ovarian cancers [17,18]. Notably, specific humoral and cellular immune responses have repeatedly been reported, and anti-NY-ESO-1 vaccination has been proposed for ovarian cancer treatment [18].

We hypothesized that targeted immunotherapies could provide innovative therapeutic options for patients with platinum-resistant HGSOC. To begin to explore this issue, in this study we assessed the expression of highly immunogenic MAGE-A10 and NY-ESO-1 CTAs, at the protein level, in advanced stage HGSOC and we investigated its correlation with responsiveness to first-line platinum-based chemotherapy.

## 2. Materials and Methods

### 2.1. Study Design

This study was based on an updated analysis of a previously investigated cohort of patients [5], focusing on 93 patients with histologically confirmed International Federation of Gynecology and Obstetrics (FIGO) III and IV stage HGSOC [19] treated at the Clinical Hospital Centre Split and General Hospital Zadar, Croatia, between January 1996 and December 2013. All patients had undergone debulking surgery followed by first line platinum-based chemotherapy. Inclusion in the study required availability of primary tumor specimens collected at initial laparotomy and full medical data.

Patients were classified according to FIGO stage [19], tumor grade [20], residual tumor after primary surgery [21], age, chemotherapy regimens, and number of cycles of chemotherapy, as previously described [5]. Response rates, progression-free (PFS) and overall survival (OS) data were obtained from histopathological reports and the patients’ medical records.

### 2.2. Chemotherapy Treatment

A large majority of the patients (*n* = 83, 89%) received paclitaxel plus platinum combinations. In particular, 81 (87%) patients received paclitaxel plus cisplatin/carboplatin (TC) every three weeks or as dose-dense (DD) TC and 2 (2%) patients received cisplatin, gemcitabine, and paclitaxel (TCG). All other patients (*n* = 10, 11%), received cisplatin-based chemotherapy without paclitaxel. In particular, 7 (8%) patients were administered cisplatin, doxorubicin, and cyclophosphamide (CAP), 2 (2%) cisplatin and cyclophosphamide (CC) and 1 (1%) patient received cisplatin only [5].

Among all the patients, 60 (65%) were administered 6 cycles of chemotherapy and 33 (35%) received more than 6 cycles of treatment. Response to platinum-based chemotherapy was defined according to the RECIST 1.1 criteria [22].

Sensitivity to treatment was defined according to the platinum-free interval (PFI) as platinum-refractory, platinum-resistant and platinum-sensitive according to standard criteria [2,5,23].

The Ethical Committee for Biomedical Research of the Clinical Hospital Split and School of Medicine approved this research to be in compliance with the Helsinki Declaration (reference number 49-1/06).

### 2.3. Immunohistochemical Staining

Immunohistochemistry (IHC) was performed as previously detailed [24] on 4 μm thick sections from paraffin embedded tissues, using as primary reagents, monoclonal antibodies (mAb) 3GA11 (anti MAGE-A10) and D8.38 (anti NY-ESO-1) [25,26] on an automated system Ventana BenchMark Ultra autostainer (Roche, Tucson, AZ, USA). Sections of normal human testes served as positive controls. Cells were considered positive if staining was detectable in either the cytoplasm or the nuclei, or both, regardless of intensity. Percentages of positive tumor cells were evaluated by two pathologists. The cut-off point for positive tumor classification was any convincing cytoplasmic/nuclear expression in >10% tumor cells.

### 2.4. Statistical Analysis

Correlations between the clinical-pathological parameters and MAGE-A10 or NY-ESO-1 positivity, as defined above, were analyzed by using Chi-squared tests. Patients survival was evaluated by using Kaplan–Meier survival curves, and differential survival was investigated by using Log-rank tests. Multivariate Cox’s proportional hazard’s analysis was used to explore the potential ability of MAGE-A10 or NY-ESO-1 protein expression to predict responsiveness to platinum-based chemotherapy and *p* values ≤0.05 in all cases were considered statistically significant. All analyses were conducted by using the SPSS version 16.0 software package.

## 3. Results

### 3.1. Demographics and Clinical-Pathological Characteristics

A total of 93 patients with advanced HGSOC were included in the study. In particular, tumors from 72 patients (77%) were of FIGO III stage and tumors from 21 patients (23%) were of FIGO IV stage. The median age of patients was 57 years (IQR: 37–79 years). The median follow-up was 60 (IQR: 4–175) months. The clinical-pathological characteristics of these patients, including FIGO stage, surgery outcome, chemotherapy regimens and number of treatment cycles, response to treatment, PFS and OS, as obtained from histopathological reports and patient medical records, are reported in Table 1.

A total of 67 patients (72%) had died and 26 (28%) were alive at the end of the follow-up (2016). Out of the alive patients, 9 (35%) had developed recurrence despite ongoing adjuvant treatment, and 17 (65%) remained without disease recurrence.

### 3.2. Immunohistochemistry

As previously reported [24], MAGE-A10 immunostaining was mainly detectable in the nuclei of malignant cells, whereas NY-ESO-1 specific mAb mainly stained cytoplasm (Figure 1). Overall, MAGE-A10 immunostaining was positive in 47 (50%) advanced HGSOCs, and NY-ESO-1 in 33 tumors (35%). Co-expression of both CTAs was observed in 23 (24.7%) cancers.

### 3.3. Response to Chemotherapy

All patients were administered cis-platinum in combination with additional chemotherapeutics in ≥6 cycles of treatment (see above). A full analysis of responsiveness to treatment has previously been reported [5]. Briefly, according to the RECIST 1.1 criteria, 62 (66.7%) patients showed a CR, 11 (11.8%) a PR, and in 5 (5.4%) patients SD represented the best response to treatment. In contrast, 15 (16.1%) patients were unresponsive and PD was evident (Table 1).

Regarding sensitivity to first-line platinum-based chemotherapy, defined as PFI, tumors from 58 (62.4%) patients appeared to be sensitive, 23 (24.7%) resistant, and 12 (12.9%) refractory. To facilitate further analyses, patients with resistant and refractory disease were combined into one group of “resistant” subjects.

The median PFS was 16 months (IQR range 4–175 months) and the median OS was 40 months (IQR range 7–175 months) [5].

### 3.4. Correlation between CTA Expression and Response to Chemotherapy

MAGE-A10 expression appeared to significantly predict unresponsiveness to first-line platinum-based chemotherapy (*p* = 0.005) and poor sensitivity to platinum treatment (*p* < 0.001) (Table 2). Accordingly, MAGE-A10 expression in tumor cells was associated with significantly poorer PFS (*p* < 0.001) and OS (*p* < 0.001) (Figure 2).

Multivariate analysis showed that, together with the patients’ age and tumor stage, MAGE-A10 expression by tumor cells was an independent predictor of unresponsiveness to first-line platinum treatment (*p* = 0.005) (Table 3).

On the same line, multivariate analysis of sensitivity to platinum, evaluated as PFI, showed that MAGE-A10 expression in tumor cells independently predicted poor sensitivity to treatment (Table 4)

Instead, no correlation was observed between NY-ESO-1 and the response to first line platinum-based chemotherapy (*p* = 0.832), platinum sensitivity (*p* = 0.168), and the patients’ PFS (*p* = 0.126) or OS (*p* = 0.335).

## 4. Discussion

Advanced ovarian cancers are characterized by poor prognosis. Following surgery, treatment mainly relies on platinum-based chemotherapy. Yet, sizeable percentages of patients do not respond to treatment, and the molecular mechanisms underlying unresponsiveness are largely unclear [2,3,4,27,28,29].

Markers associated with sensitivity of ovarian cancer cells to treatment have been identified by us and others. In particular, E cadherin downregulation and overexpression of epithelial-to-mesenchymal transition (EMT) genes are known features of unresponsive tumors [5,29]. Accordingly, enhanced tumor cell de-differentiation is known to underlie resistance to different types of chemotherapy in a variety of malignancies [30], thus suggesting that treatments based on the administration of conventional anti-cancer compounds might be poorly effective.

On the other hand, notably, immunotherapy based on immunological checkpoint blockade also appears to be largely ineffective in ovarian cancers.

Innovative approaches are urgently needed.

CTAs are a large family of tumor-associated antigens expressed in healthy germ cells and in a variety of human cancers [11,12,13,14]. As such, they do represent attractive candidates for cancer immunotherapy. In particular, MAGE-A10 and NY-ESO-1 are highly immunogenic CTAs and specific immune responses have repeatedly been observed in patients bearing cancers expressing these antigens [14,15,16,17].

The physiological functions of CTAs are still largely unclear. Data on MAGE-A, the best studied CTAs, suggests that they are involved in early oncogenesis, and in the regulation of cell cycle progression and cellular apoptosis [10]. Expression of MAGE-A and NY-ESO-1 proteins has previously been shown to be associated with poor prognosis by us and others in a variety of cancers of different histological origin [11,12,13,14,31]. Moreover, interestingly, overexpression of CTA genes has also been suggested to correlate with resistance to chemotherapy in head and neck cancers and in medulloblastoma [32,33,34,35,36]. Intriguingly, defined established tumor cell lines expressing MAGE-A CTA have been reported to be platinum resistant [36].

Underlying mechanisms have been explored and evidence has been provided suggesting that the expression of MAGE-A genes inhibits cancer cell apoptosis, possibly by modulating wild type TP53 gene expression [33,37,38]. On the other hand, MAGE-A CTA gene expression might represent a marker of widespread DNA de-methylation, critically contributing to the selection of chemoresistant malignant cell subclones [39].

Regarding ovarian cancers, expression of MAGE-A4, MAGE-A9, MAGE-A10, and NY-ESO-1 proteins in tumor cells has previously been reported to be associated with poor prognosis [14,40,41,42], but no data regarding sensitivity to chemotherapy treatment have been provided.

Here we report that MAGE-A10 protein expression in ovarian cancer surgical specimens reliably predicts unresponsiveness to first line platinum-based chemotherapy and poor sensitivity to platinum treatment. Accordingly, consistent with previous reports [14], MAGE-A10 protein expression is associated with significantly poorer PFS and OS. Most importantly, multivariate analysis of our data indicates that MAGE-A10 protein expression qualifies as predictor of resistance to treatment independently from age and FIGO stage.

In contrast, we did not observe any correlation between NY-ESO-1 protein expression and response to first-line platinum-based chemotherapy, platinum sensitivity, or the patients’ PFS and OS.

To the best of our knowledge, this is the first study evaluating the impact of MAGE-A10 protein expression on HGSOC chemosensitivity. Therefore, MAGE-A10 could represent a novel biomarker identifying a more aggressive phenotype. Further analysis could be undertaken in patients with secondary cytoreduction and contribute to clinical decisions regarding the implementation of radical surgery procedures [43].

The limitations of our report should be acknowledged. They include a small number of patients and a relatively short follow-up time. Nevertheless, although validation and mechanistic studies are warranted, our results could be of high clinical relevance. First, assessment of MAGE-A10 protein expression might help to identify patients with platinum-resistant tumors, thereby sparing them chemotherapy-associated adverse effects. On the other hand, considering the high immunogenicity of MAGE-A10, it is tempting to speculate that immunotherapy specifically targeting this CTA, based on vaccination or adoptive immunotherapy with engineered T cells expressing a MAGE-A10-specific HLA-restricted T-cell receptor [44] might powerfully complement platinum-based chemotherapy, by eliminating resistant cells.

Cellular and humoral immune responses specific for cancer/testis antigens (CTAs), and, in particular, MAGE-A10, have frequently been observed in patients with tumors of different histological origin (e.g., [14,15,16]) and were not reported to be associated with clinical symptoms suggesting ongoing autoimmune reactions targeting healthy tissues. Most notably, human spermatogonia do not express HLA Class I determinants [45], and in females MAGE-A CTA expression is only detectable in the placenta [46,47]. Moreover, vaccination against MAGE-A CTA, irrespective of vaccine formulation and adjuvant has not been shown to be associated with on-target off-tumor toxicity [48,49,50,51].

However, adoptive treatment with T cells transduced with genes encoding a HLA-A2-restricted MAGE-A3 specific T-cell receptor (TCR) has been shown to result in severe neurotoxicity, initially attributed to the recognition of rare MAGE-A12-expressing neurons [52], and, more recently, to the targeting of EPS8L2 protein expressed in multiple tissues [53]. Instead, a similar adoptive treatment targeting MAGE-A4 and including patients with ovarian cancer has successfully been implemented [48,49,50,51], with one patient developing pseudogout arthritis [54,55]. Adoptive treatments with HLA-Class II restricted transduced CD4^+^ T cells have also been proposed [56].

On the other hand, MAGE-A10 has only been targeted in three clinical trials based on peptide vaccination [51,56,57]. Moreover a MAGE-A10 targeted adoptive immunotherapy protocol has recently been used in patients with advanced NSCLC with an acceptable safety profile [58].

Thus, while similar therapeutic approaches should be considered cautiously, intraperitoneal administration of T cells engineered to express a MAGE-A10-specific HLA-restricted T-cell receptor might prove of particular interest in cases of ovarian cancer-associated peritoneal carcinomatosis [59], typically characterized by a paucity of therapeutic options.

## 5. Conclusions

Molecular assessment of MAGE-A family members could be helpful to improve the prognostic evaluation and to provide a new potential therapeutic target for epithelial ovarian cancer patients.

## Figures and Tables

**Figure 1 cancers-15-04697-f001:**
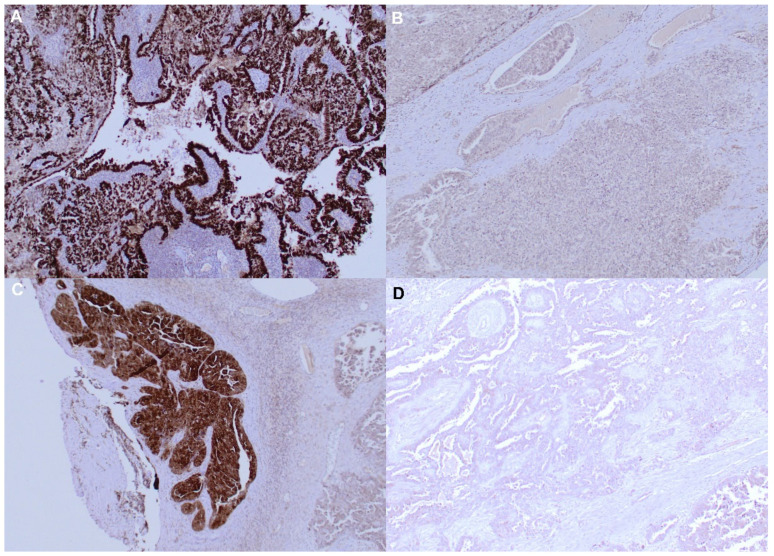
MAGE-A10 and NY ESO protein detection by IHC staining in ovarian tumor samples. (**A**) Positive MAGE-A10-specific IHC staining; (**B**) Negative MAGE-A10-specific IHC staining; (**C**) Positive NY ESO-specific IHC staining; (**D**) Negative NY ESO-specific IHC staining. Original magnification ×200.

**Figure 2 cancers-15-04697-f002:**
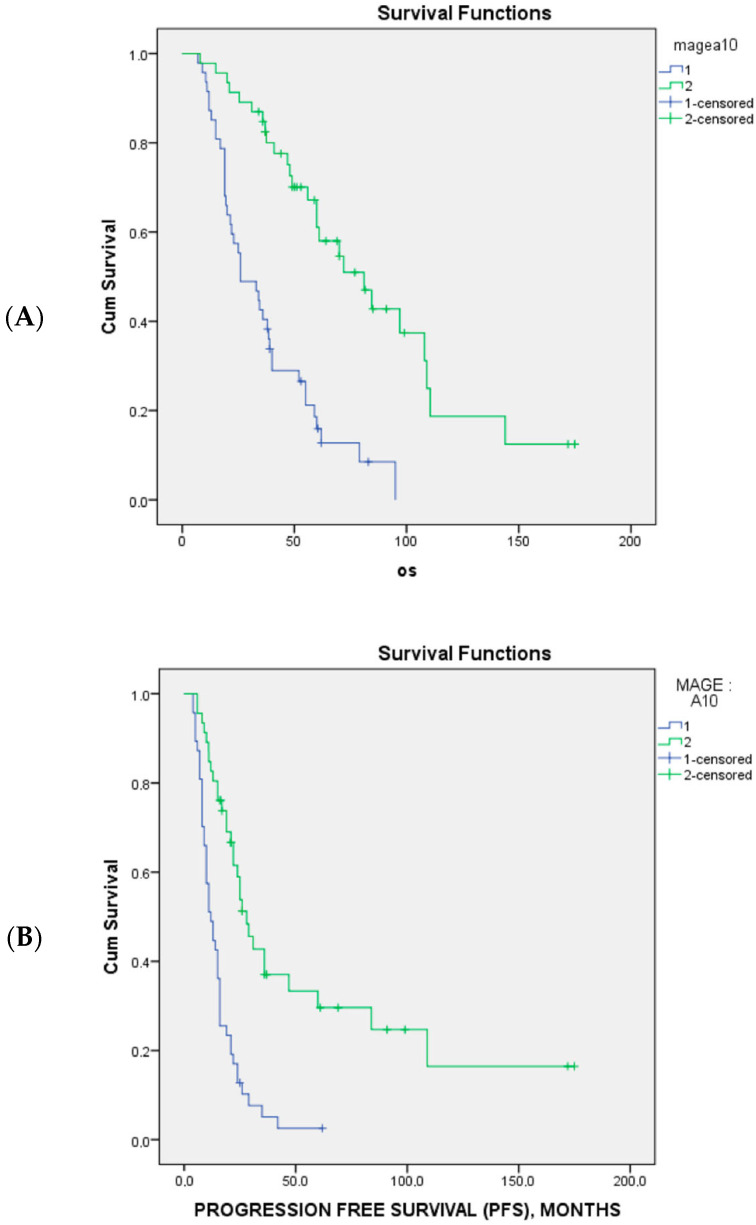
The Kaplan–Meier survival curves of overall survival (**A**) and progression-free survival (**B**) and MAGE A10 expression in 94 patients with advanced-stage high-grade serous ovarian cancer. The blue line is MAGE 10 positive and green line are MAGE 10 negative.

**Table 1 cancers-15-04697-t001:** Demographics and clinical-pathological characteristics.

		Disease Status at the End of Follow-Up			
		Total	Dead	Alive	Alive
		(N = 93)	(N = 67)	with Recurrence (N = 9)	without Recurrence (N = 17)
Age (median, IQR)		57(37–79)	59(37–79)	62(49–75)	54(44–76)
PFS (median, IQR)		16(4–175)	14(4–109)	22(8–42)	37(16–175)
OS (median, IQR)		40(7–175)	36(7–144)	44(34–83)	69(37–175)
FIGO stage (N, %)	III	72(77)	50(75)	6	16(94)
	IV	21(23)	17(25)	3	1(6)
Surgery (N, %)	Optimal	14(15)	0	4	10(15)
	Suboptimal	75(81)	9	12	54(80)
	Unknown	4(4)	0	1	3(4)
Chemotherapy cycles (N, %)	6	60(64)	5	16	39(58)
	>6	33(36)	4	1	28(42)
Response to chemotherapy (N, %)	CR	62(66,7)	40(59,7)	6	16
	PR	11(11,8)	8(11,9)	2	1
	SD	5(5,4)	5(7,5)	0	0
	PD	15(16,1)	14(20,9)	1	0

N: number; PFS: progression-free survival; OS: overall survival; CR: complete response; PR: partial response; SD: stable disease; PD: progressing disease.

**Table 2 cancers-15-04697-t002:** Correlation between CTA expression and response to chemotherapy.

	N (%)				
		Response to Chemotherapy			
		CR + PR = OR	SD + PD	χ²	*p*
MAGE A-10	Positive	30(41)	17(85)	10.4	0.001
	Negative	43(59)	3(15)		
NY-ESO-1	Positive	25(34)	8(40)	0.045	0.832
	Negative	48(66)	12(60)		
Chemotherapy sensitivity	Sensitive	58(80)	0	39	<0.001
	Resistant	15(20)	20(100)		

N: number; CR: complete response; PR: partial response; OR: overall response; SD: stable disease; PD: progressing disease.

**Table 3 cancers-15-04697-t003:** Multivariate analysis of the age of patients, FIGO stage and expression of MAGE A-10 on objective response on first-line chemotherapy in 93 patients with advanced-stage high-grade serous ovarian cancer.

	HR (95%CI)	*p*
Age	4.2 (1.2–14.6)	0.025
Stage	5.7 (1.6–21)	0.007
MAGE A10	7.4 (1.8–29.8)	0.005

**Table 4 cancers-15-04697-t004:** Multivariate analysis of the impact of FIGO stage, chemotherapy cycles, and expression of MAGE A-10 on chemotherapy sensitivity in 93 patients with advanced-stage HGSOC.

	HR (CI 95%)	*p*
Stage	3.1 (1.0–9.9)	0.050
MAGE A10	6 (2.2–16.4)	<0.001
Chemotherapy cycles	1.8 (0.65–4.8)	0.265

## Data Availability

Data available upon request from the corresponding author.

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
