# Peer review of "MAGE-A10 Protein Expression in Advanced High Grade Serous Ovarian Cancer Is Associated with Resistance to First-Line Platinum-Based Chemotherapy"

_cancers, 2023, doi:10.3390/cancers15194697_

Round 1
Reviewer 1 Report
Nataša LisicaŠikić and collaborators investigate the expression level of two cancer/testis antigens (MAGE-A10 and NY-ESO-1) in human specimens of advanced-stage high-grade serous ovarian carcinoma and correlate their expression level to the clinical-pathological parameters. The authors demonstrate that MAGE-A10 protein expression is independent predictor of unresponsiveness to platinum treatment and shorter overall survival.
Although the manuscript is interesting, it cannot be accepted in its current form for publication on Cancers. Issues to be addressed are the following:
1 The authors stated that MAGE-A10 expression in tumor cells was associated with significantly poorer PFS (p<0.001) and OS (p<0.001) (lines 195-196). The figure 2 shows the association between the expression of MAGE-A10 and the OS, no data are shown on the association between the expression of MAGE-A10 and the PFS. Please include in figure 2 the graph showing the association between the expression of MAGE-A10 and the PFS. In addition, in the legend it should be described which line represents the MAGE-A10 high expression group and which line represents the MAGE-A10 low expression group.
2 Could the expression of CTA and in particular of MAGE-A10 in healthy germ cells limit the use of immunotherapy in ovarian cancer treatment? How the cancer immunotherapy based on MAGE-A10/CTA could damage the healthy germ cells? These issues should be addressed and discussed in the discussion section.
3 The authors should present and discuss in more detail the antitumoral vaccination and/or adoptive immunotherapy preclinical studies (if available) and clinical trials (carried out not only on ovarian cancer patients but also on patients affected by different cancer types) with MAGE-A10 presenting the benefits and limitations. A table summarizing the results of clinical trials could be helpful and informative.
Minor:
Table 1: the total number of patients showing the partial response is wrong. It is 11 not 1. Please check.
Table 1: in table 1 two groups of chemotherapeutic teatment are shown: Chemotherapy cycles < 6 and Chemotherapy cycles > 6. In the text, lines 180-181, the authors stated that all patients were administrated the chemotherapeutic agents in ≥6 cycles of treatment. Please check the sentences and the table.
Table 2: please include the meaning of OR in the footnotes of table 2 and remove the meaning of PFS and OS whose data are not shown in table 2
Line 236: check the sentence “Physiological functions of CTA are by and large still unclear”
Line 256: replace “ins” with “in”
The manuscript is clearly written. Moderate editing of English language is required.
Author Response
Thank you very much for taking the time to review this manuscript. Please find the detailed responses below and the corresponding revisions/corrections highlighted/in track changes in the re-submitted files
Dear Reviewer,
Thanks for reviewing our study entitled “MAGE-A10 protein expression in advanced high grade serous ovarian cancer is associated with resistance to first line platinum-based chemotherapy” recently submitted to your attention for publication in “Cancers”,
The paper has been revised to comply with reviewers’ criticisms.
In particular, regarding comments from Reviewer 1:
1 The authors stated that MAGE-A10 expression in tumor cells was associated with significantly poorer PFS (p<0.001) and OS (p<0.001) (lines 195-196). The figure 2 shows the association between the expression of MAGE-A10 and the OS, no data are shown on the association between the expression of MAGE-A10 and the PFS. Please include in figure 2 the graph showing the association between the expression of MAGE-A10 and the PFS. In addition, in the legend it should be described which line represents the MAGE-A10 high expression group and which line represents the MAGE-A10 low expression group.
The figure has been amended as required.
2 Could the expression of CTA and in particular of MAGE-A10 in healthy germ cells limit the use of immunotherapy in ovarian cancer treatment? How the cancer immunotherapy based on MAGE-A10/CTA could damage the healthy germ cells? These issues should be addressed and discussed in the discussion section.
3 The authors should present and discuss in more detail the antitumoral vaccination and/or adoptive immunotherapy preclinical studies (if available) and clinical trials (carried out not only on ovarian cancer patients but also on patients affected by different cancer types) with MAGE-A10 presenting the benefits and limitations. A table summarizing the results of clinical trials could be helpful and informative.
Cellular and humoral immune responses specific for Cancer/Testis antigens (CTA), and, in particular, MAGE-A10, have frequently been observed in patients with tumors of different histological origin (e.g. 1-4) and were not reported to be associated with clinical symptoms suggesting ongoing autoimmune reactions targeting healthy tissues. Most notably, human spermatogonia do not express HLA Class I determinants (5), and in females MAGE-A CTA expression is only detectable in placenta (6-7). Moreover, vaccination against MAGE-A CTA, irrespective of vaccine formulation and adjuvant has not been shown to be associated with on-target off-tumor toxicity (8-11).
However, adoptive treatment with T cells transduced with genes encoding a HLA-A2-restricted MAGE-A3 specific T-cell receptor (TCR) has been shown to result in severe neurotoxicity, initially attributed to recognition of rare MAGE-A12 expressing neurons (12), and, more recently, to targeting of EPS8L2 protein expressed in multiple tissues (13). Instead, a similar adoptive treatment targeting MAGE-A4 and including patients with ovarian cancer has successfully been implemented (14), with one patient developing a pseudogout arthritis (15). Adoptive treatments with HLA-Class II restricted transduced CD4+ T cells have also been proposed (16).
On the other hand, MAGE-A10 has only been targeted in three clinical trials based on peptide vaccination (11, 16, 17). Moreover a MAGE-A10 targeted adoptive immunotherapy protocol has recently been used in patients with advanced NSCLC with an acceptable safety profile (19).
While these studies are cited in the discussion, they do not obviously appear to provide sufficient material for a dedicated table. Remarkably, a recent article published in “Cancers” a few months ago nicely reviews previous and ongoing clinical trials targeting MAGE-A CTA (20). This review is now cited in the “Discussion” section of our revised paper.
This information is now included in the main text, in the “Discussion” section (lines 277-300). Bibliography has been revised accordingly.
Minor:
Table 1: the total number of patients showing the partial response is wrong. It is 11 not 1. Please check.
Table 1: in table 1 two groups of chemotherapeutic teatment are shown: Chemotherapy cycles < 6 and Chemotherapy cycles > 6. In the text, lines 180-181, the authors stated that all patients were administrated the chemotherapeutic agents in ≥6 cycles of treatment. Please check the sentences and the table.
Table 2: please include the meaning of OR in the footnotes of table 2 and remove the meaning of PFS and OS whose data are not shown in table 2
Line 236: check the sentence “Physiological functions of CTA are by and large still unclear”
Line 256: replace “ins” with “in”
Minor comments were addressed, as indicated.
Reviewer 2 Report
Dear Editor's and Anthors,
I have completed the review of the manuscript titled "MAGE-A10 protein expression in advanced high-grade serous ovarian cancer is associated with resistance to first-line platinum-based chemotherapy."
The paper investigates the role of MAGE-A10 and NY-ESO-1 protein expression in predicting the response to platinum-based chemotherapy in advanced high-grade serous ovarian carcinoma (HGSOC). The study analyzes data from 93 patients treated between 1996 and 2013, finding a significant association between MAGE-A10 expression and resistance to platinum-based chemotherapy, poor platinum sensitivity, and reduced PFS and OS. In contrast, NY-ESO-1 protein expression does not correlate with treatment response. The authors suggest that MAGE-A10 could serve as a biomarker for identifying patients with platinum-resistant tumors, potentially guiding treatment decisions.
While your findings are promising, I have some suggestions and concerns regarding the results of your manuscript.
1) While the study is informative, it has certain limitations such as a small sample size. One important step in establishing the clinical utility of MAGE-A10 as a predictive biomarker for chemotherapy resistance is validating your findings in an independent cohort of patients. This validation is essential to confirm the robustness and generalizability of your results. I recommend conducting further studies with a separate patient cohort to verify the association between MAGE-A10 expression and chemotherapy resistance in HGSOC. If it is not feasible to conduct validation in another cohort, it is crucial that you provide a clear and well-justified explanation for this limitation in your manuscript. Authors should discuss the practical challenges or constraints that prevented them from conducting validation, such as limited access to patient samples, resource constraints, or other logistical issues. Additionally, consider discussing directions for future research, including potential strategies for overcoming these limitations?
2) While the paper highlights the association between MAGE-A10 expression and chemotherapy resistance, it does not delve into the underlying molecular mechanisms. To provide a more comprehensive understanding, could the authors discuss potential mechanisms by which MAGE-A10 might contribute to platinum resistance in HGSOC?
3) In scientific research, the clarity and visual impact of figures are crucial for conveying research findings and insights effectively. To enhance the clarity of your study, I would like to suggest that you consider using professional graphing software such as GraphPad Prism to redraw Figure 2.
Please convey these recommendations to the authors and request that they address them comprehensively in a revised version of the manuscript. Thank you for considering my review, and I look forward to the authors' response and the revised manuscript.
The use of academic language is generally good. However, ensure that the language is consistently formal and appropriate for a scientific publication. Here are some comments on the language and grammar issues in the manuscript:
Abstract:
In the abstract, consider rephrasing "unresponsive ones treatment-associated toxicity" to "patients unresponsive to treatment-associated toxicity."
Introduction:
In the introduction, you can enhance clarity by specifying what "FIGO" stands for before using the abbreviation. For example, "International Federation of Gynecology and Obstetrics (FIGO) stage."
In the sentence, "A multiplicity of molecular mechanisms have been suggested," it may be clearer to say, "Multiple molecular mechanisms have been proposed."
Methods:
Instead of "In all cases ≤0.05 p values," use "p values ≤0.05 in all cases."
In the sentence, "Median age of patients was 57 years," consider specifying the range or interquartile range for better clarity. For example, "Median age of patients was 57 years (IQR: 37-79 years)."
Please ensure that these revisions are made to improve the clarity and readability of the manuscript.
Author Response
Dear Reviewer,
Thank you for your time reading our manuscript and giving valuable suggestions. Please find our comments to your questions.
- While the study is informative, it has certain limitations such as a small sample size. One important step in establishing the clinical utility of MAGE-A10 as a predictive biomarker for chemotherapy resistance is validating your findings in an independent cohort of patients. This validation is essential to confirm the robustness and generalizability of your results. I recommend conducting further studies with a separate patient cohort to verify the association between MAGE-A10 expression and chemotherapy resistance in HGSOC. If it is not feasible to conduct validation in another cohort, it is crucial that you provide a clear and well-justified explanation for this limitation in your manuscript. Authors should discuss the practical challenges or constraints that prevented them from conducting validation, such as limited access to patient samples, resource constraints, or other logistical issues. Additionally, consider discussing directions for future research, including potential strategies for overcoming these limitations?
We fully agree with this comment and limitations of our study are clearly stated in the text (lines 267-270). However, limited access to non-commercially available reagents and clinical specimens do preclude the performance of a validation study complementing the present report, although future additional research is warranted. It should also be noted that although correlation between chemotherapy resistance and expression of other MAGE-A CTA at the gene level was previously reported (21-25), this is the first study addressing it for MAGE-A10, and at the protein level. Therefore, while its limitations are clearly acknowledged, its ice-breaking nature, paving the way for additional studies should be appreciated.
- While the paper highlights the association between MAGE-A10 expression and chemotherapy resistance, it does not delve into the underlying molecular mechanisms. To provide a more comprehensive understanding, could the authors discuss potential mechanisms by which MAGE-A10 might contribute to platinum resistance in HGSOC?
Data from previous studies based on the analysis of established cell lines suggest that expression of MAGE-A genes inhibits cancer cell apoptosis, possibly by modulating wild type TP53 gene expression (22, 26, 27). On the other hand, MAGE-A CTA gene expression might represent a marker of widespread DNA de-methylation, critically contributing to the selection of chemoresistant malignant cell subclones (28) . This information has now been included in the “Discussion” section of our study lines (244-248), as recommended, and bibliography has been updated.
) In scientific research, the clarity and visual impact of figures are crucial for conveying research findings and insights effectively. To enhance the clarity of your study, I would like to suggest that you consider using professional graphing software such as GraphPad Prism to redraw Figure 2.
The figure has been restructured, as required.
Abstract:
In the abstract, consider rephrasing "unresponsive ones treatment-associated toxicity" to "patients unresponsive to treatment-associated toxicity."
Introduction:
In the introduction, you can enhance clarity by specifying what "FIGO" stands for before using the abbreviation. For example, "International Federation of Gynecology and Obstetrics (FIGO) stage."
In the sentence, "A multiplicity of molecular mechanisms have been suggested," it may be clearer to say, "Multiple molecular mechanisms have been proposed."
Methods:
Instead of "In all cases ≤0.05 p values," use "p values ≤0.05 in all cases."
In the sentence, "Median age of patients was 57 years," consider specifying the range or interquartile range for better clarity. For example, "Median age of patients was 57 years (IQR: 37-79 years)."
Text has been corrected as required.
Hoping that the manuscript, as presently modified, matches Reviewers’ and Editor’s requirements,
and looking forward to hearing from you at your convenience,
I remain,
Sincerely,
Luka Matak
Round 2
Reviewer 1 Report
Minor revision:
Line 24: please remove “a” from “as a biomarkers”
Line 25: check the sentence “because a subset of the proteins that are classified as cancer-testis antigens (CTA).”
The manuscript is clearly written. Minor editing of English language is required (see comments for authors).
Reviewer 2 Report
The authors have addressed most of the concerns and suggestions raised during the review process, resulting in a more coherent and well-organized manuscript. The manuscript is now suitable for publication.